# Diagnosis of Pneumonia by Cough Sounds Analyzed with Statistical Features and AI

**DOI:** 10.3390/s21217036

**Published:** 2021-10-23

**Authors:** Youngbeen Chung, Jie Jin, Hyun In Jo, Hyun Lee, Sang-Heon Kim, Sung Jun Chung, Ho Joo Yoon, Junhong Park, Jin Yong Jeon

**Affiliations:** 1Department of Mechanical Engineering, Hanyang University, 222 Wangsimri-ro, Seongdong-gu, Seoul 04763, Korea; chung_911@naver.com; 2School of Electromechanical and Automotive Engineering, Yantai University, 30 Qingquan Road, Laishan District, Yantai 264005, China; jinjie910@sina.com; 3Department of Architectural Engineering, Hanyang University, 222 Wangsimri-ro, Seongdong-gu, Seoul 04763, Korea; best2012@naver.com; 4Department of Internal Medicine, Hanyang University Hospital, Hanyang University College of Medicine, 222 Wangsimri-ro, Seongdong-gu, Seoul 04763, Korea; namuhanayeyo@hanyang.ac.kr (H.L.); wisesung82@hanmail.net (S.J.C.); hjyoon@hanyang.ac.kr (H.J.Y.); 5Department of Medical and Digital Engineering, Hanyang University, 222 Wangsimri-ro, Seongdong-gu, Seoul 04763, Korea; jyjeon@hanyang.ac.kr

**Keywords:** cough, pneumonia, machine-learning, artificial intelligence, long short-term memory, loudness, energy ratio

## Abstract

Pneumonia is a serious disease often accompanied by complications, sometimes leading to death. Unfortunately, diagnosis of pneumonia is frequently delayed until physical and radiologic examinations are performed. Diagnosing pneumonia with cough sounds would be advantageous as a non-invasive test that could be performed outside a hospital. We aimed to develop an artificial intelligence (AI)-based pneumonia diagnostic algorithm. We collected cough sounds from thirty adult patients with pneumonia or the other causative diseases of cough. To quantify the cough sounds, loudness and energy ratio were used to represent the level and its spectral variations. These two features were used for constructing the diagnostic algorithm. To estimate the performance of developed algorithm, we assessed the diagnostic accuracy by comparing with the diagnosis by pulmonologists based on cough sound alone. The algorithm showed 90.0% sensitivity, 78.6% specificity and 84.9% overall accuracy for the 70 cases of cough sound in pneumonia group and 56 cases in non-pneumonia group. For same cases, pulmonologists correctly diagnosed the cough sounds with 56.4% accuracy. These findings showed that the proposed AI algorithm has value as an effective assistant technology to diagnose adult pneumonia patients with significant reliability.

## 1. Introduction

Pneumonia is a serious condition that is often accompanied by complications, which sometimes lead to death [1]. Notwithstanding most cases of pneumonia-related deaths being preventable by timely diagnosis and cost-effective treatment, the disease is often plagued by delayed diagnoses [2]. Diagnosis is usually made based on clinical features, chest examinations, plain chest X-rays, or chest computerized tomography findings [3]. However, under special circumstances and in areas having poor access to proper medical resources, early detection of pneumonia is difficult if not impossible. Moreover, the global 2019 novel coronavirus (COVID-19) pandemic has demonstrated the need for new diagnostic tools that minimize medical personnel engagement while avoiding equipment being exposed to afflicted patients [4].

Traditionally, a cough is not considered a disease-specific symptom, except in unique situations. However, recent advancements in artificial intelligence (AI) techniques have led to various studies that have attempted to classify respiratory diseases by cough sounds, including pediatric pneumonia [5], asthma [6,7,8,9], croup [10], pertussis [11,12] and acute bronchitis [13]. The deep-learning system of Porter et al. [14] classified cough sounds from 585 children for five diseases (asthma, pneumonia, bronchitis, croup and acute lower respiratory tract infection), showing an accuracy ≥80%. Abeyratne et al. [15,16] extracted mel-frequency cepstral coefficient (MFCC) and non-Gaussian acoustic characteristics for coughs caused by pneumonia, bronchitis and asthma for 91 pediatric patients, using a logistic regression model (LRM)-based classification system to achieve 94% sensitivity and 63% specificity for diagnosis. Hee et al. [6] extracted MFCC and constant Q-transform characteristics to classify asthma and normal coughs. They recorded 1192 coughs from 89 children using a Gaussian mixture model-universal background model, showing 83% sensitivity and 85% specificity. Sharan et al. [10,17] extracted MFCC and cochleagram image features for diagnosing croup in 479 children and proposed an LRM and a support vector machine (SVM)-based classification system, with which, 92% sensitivity and 85% specificity were achieved. Botha [18] used body temperature, BMI and heart rate to analyze the MFCCs of cough sounds in 38 children for diagnosis of pulmonary tuberculosis. By using an HMM-based classification system, results of 95% sensitivity and 72% specificity were achieved. These previous studies have been pioneering roles by combining cough sound signals and AI technology to diagnose respiratory disease of pediatric patients. However, analysis of cough sounds for the diagnosis of pneumonia in adults has not been reported yet.

Apart from the active development of AI-based systems for diagnosing respiratory diseases, the acoustic features of coughs used in algorithms relied on spectral parameters calculated used for random sounds. Thus, it was required to find the unique statistical features of cough sounds to interpret diagnostic system performance. The cough sounds were collected from patients. Their temporal and spectral characteristics were analyzed to find the representative features depending on the symptoms. With these features, this study presents distinction of pneumonia in adult patients for the development of an AI-based pneumonia diagnostic algorithm. We also assessed its performance in the diagnosis of pneumonia in adult cough patients. The following section materials and methods describes in detail the patient groups who participate in the study, how to collect and acoustically analyze cough sounds, construct AI algorithm, and evaluate their performance. And the results section mainly analyzes the study population, the characteristic of cough sounds from pneumonia and non-pneumonia groups, and the pneumonia diagnosis results using developed algorithm. Finally, discussion section describes the meaning and value of the results found through this study. And considers the limitations so that it can be further developed through subsequent research.

## 2. Materials and Methods

### 2.1. Study Population and Clinical Diagnosis of Causes of Coughs

The study enrolled adult patients older than 20 years who visited a pulmonary department of the university hospital for complaints of acute or chronic cough. Drug-induced cough and irritant-induced cough by smoking or occupational exposure were excluded.

The present study was approved by the institutional review board (No. 2017-02-019). The protocol was performed in accordance with the relevant guidelines and regulations. The informed written consent was also received from each participant. We collected the demographic and clinical information including cough durations, accompanying respiratory symptoms, and smoking status. Cough severity was assessed using a 10-cm horizontal visual analogue scale by the subjects. Diagnoses of causative diseases were made by respiratory specialists based on patients’ clinical symptoms, physical examination, and imaging studies. All the patients underwent chest X-ray imaging study, while only those with rhinitis or sinusitis symptoms received paranasal sinus imaging tests.

In the evaluation of non-pneumonia causes of cough, pulmonary function and methacholine provocation tests were performed based on the decision by the pulmonologists in cases suggestive of asthma, chronic obstructive pulmonary disease, or eosinophilic bronchitis. Cases of pneumonia were defined as those having clinical features suggesting pneumonia and pulmonary infiltration or consolidation lesions corresponding to pneumonia via chest imaging, whose symptoms and lung lesions improved after antibiotic therapy based on the guideline of community-acquired pneumonia [19].

### 2.2. Measurement of Cough Sounds

A portable recorder (HEAD acoustics GmbH, SQobold) and a microphone (PCB Piezotronics, 378B02) were used to collect cough sounds (Table 1). The microphone was placed 40 cm away from the mouth of the seated participant. Recording was performed while interviewing individuals with the same case report form (CRF), and in the process, each patient generated only spontaneous coughs. In the recorded file after the interview, the section in which a single cough occurred from the patient was checked. Through the pre-processing, only the single cough sound that does not include any noise like human voice or other ambient sound was selected. Finally, 126 single cough sounds were obtained from 30 patients (70 cases of single cough sound from 15 pneumonia patients; 56 cases from non-pneumonia patients). An average of 3 to 5 coughs were obtained per patient. The progress of a single cough was recognized in three phases: onset of cough; middle stage of air discharging; vocal cord closure [20,21,22].

### 2.3. Extraction of Acoustic Features of Cough Sounds

To compare the physical characteristics of the cough sounds from pneumonia and non-pneumonia groups for objective quantification, acoustic features were examined using time, frequency, psychoacoustics, and energy aspects. Preferentially, normalization for cough intensity was performed by dividing each original cough sound by the maximum value of the corresponding data. Through this normalization process for each cough sound, the influence of the intensity of the cough itself was excluded.

To quantify the acoustic features, loudness was analyzed to find the transient variation of psychoacoustic aspects. Loudness is a sound quality parameter that represents the physical sound intensity in psychoacoustics and is derived according to the calculation method specified in ISO 532B. Because loudness reflects the sound intensity, frequency, and duration comprehensively, it is sensitive to high-frequency spectral components, and therefore, it is judged to be appropriate for cough sound analysis and relevant quantification [23,24].

The energy ratio was defined and analyzed via the short-time Fourier transform (STFT) to select a reference frequency for differentiating pneumonia and non-pneumonia cases. Moreover, an empirical-mode decomposition (EMD) method was used to separate cough sounds into various intrinsic mode-function (IMF) components according to given frequency bands (Appendix A). EMD is a method of decomposing based on a raw signal without defining a parameter value with analyzing the time series signal. Therefore, it is a general-purpose signal processing method that can be applied to acoustic and vibration data including cough sounds [25,26]. Consequently, each IMF was divided into high- and low-frequency components based on the reference frequency. The ratio of summation of the components was calculated as
(1)$$R=20\log_{10}\left(\frac{H_{rms}}{L_{rms}}\right),$$
where R represent the newly defined energy ratio. Here, a single value for loudness and the energy ratio was derived from the frame at 0.05 s (starting point of cough and calculation), which was carried out while moving the frame to overlap 75% to derive characteristics according to cough duration [27,28]. $H_{rms}$ is the root mean-square (RMS) value of the high-frequency component and $L_{rms}$ is the RMS value of the low-frequency component. The loudness derived as the change in sound intensity. The energy ratio represents the energy distribution ratio between the high- and low-frequency components separated through the EMD method. These objective quantities were used as the input parameters of the diagnosis model.

### 2.4. AI-Based Diagnostic Algorithm

To establish the machine-learning algorithm, eight patients (four pneumonia patients and four non-pneumonia patients) were randomly selected, and loudness and energy ratio from their cough sounds were derived. To acquire the general pattern of change for each cough sound feature, a moving average was applied for smoothing, and the results were used as reference data. Subsequently, a training dataset was obtained via data augmentation [29,30]. This augmentation method mainly consists of two steps. The first step is distortion of the time axis. In this step, with the different value of constant, *α*, the total duration of cough sound data, *t*′, was adjusted as
(2)$$t^{\prime }=t\cdot \left(1+\frac{\alpha -1}{10}\right),$$
where *t* is the duration before variation. This augmentation was performed to minimize the influence of cough sound duration on the diagnosis.

Subsequently, the cough sounds were augmented after adjustment of the cough sound pattern by minor distortion on the timing as
(3)$$t_{a}=t^{\prime }+C\cdot \left(\frac{\beta -5}{5}\right)\cdot \sin\left(\frac{t}{t_{\max}}\cdot 4\pi \beta \right),$$
where $t_{a}$ represents the augmented cough sound occurrence instances, and *β* is a regulating constant. To take into account the difference in the absolute value of each feature, the value of proportional constant, *C*, was set to 0.02 for loudness and to 0.01 for the energy ratio augmentations.

In addition, sound amplitude was varied for additional augmentations. This process is the second step with proportional transformation of y-axis. In this step, the amplitude was changed by multiplying the value of each characteristic factor by a proportional constant. This process was effective for augmenting the collected cough sounds dataset. Because loudness and energy ratio are the time-series characteristic factors that designed to reflect the characteristics of pneumonia based on the result of the acoustic analysis for cough sounds. Using the augmentation process that involved the transformation of each data axis, data sets reflecting the paternal changes of cough sounds were generated. Simultaneously, the diversity of the total length or rate of change in cough sounds for the training dataset was achieved. To prevent overfitting, the values of *α* and *β* were set to integers between 1 and 10. The amplitudes were varied in logarithm scales between 0.8 and 1.2 [29,31]. Ultimately, a total of 43,000 training data were generated and used for the reference-based training.

In this reference-based training, AI technology based on training dataset obtained after completing the data augmentation process was applied. Various studies using AI have been conducted to perform sound event detection [32,33]. To classify pneumonia and non-pneumonia symptoms through supervised learning, the machine-learning algorithm based on long short-term memory (LSTM) was used. LSTM is a kind of RNN with forget gate added to overcome the vanishing gradient problem. LSTM trains the pattern of transient values that change according to time series [34]. Since it also learns even if the length of input factor varies, it was judged to be appropriate for cough sound-based diagnosis.

As shown in Figure 1, it was constructed to have five layers using the loudness and energy ratio of cough sounds. Sequence data is input to the algorithm through the sequence input layer. And then, based on the time series and sequence data, LSTM layer train long-term dependencies between time steps through input weights, recurrent weights, and biases. In a fully connected layer, all included neurons are connected to all neurons of the previous layer to perform classification by combining features trained from the previous layer. In the softmax layer, the softmax function is applied to the input value. Finally, the classification output layer assigned the data to one of the mutually exclusive classes by calculating the cross-entropy loss through the value obtained from softmax function [35,36]. When the minibatch size was set to 256 and the max epoch was set to 8, a training accuracy of 100% was confirmed. The performance of developed algorithm is evaluated by comparison to decision from pulmonologists based on various clinical tests. If the result of algorithm coincides with pulmonologists, that cough case is counted as true, and if it does not match, it is counted as false. Ultimately, pneumonia diagnostic performance was tested by the sensitivity, specificity, and accuracy. Also, a logistic regression model was constructed using two vectors as input variables. First vector represents the diagnostic results by actual clinical tests and the other is decisions based on only cough sounds. The receiver operating characteristic (ROC) curve of developed algorithm and each pulmonologist was plotted from the statistical model, and quantitative comparison was performed through the area under the curve (AUC) [37].

### 2.5. Pneumonia Diagnosed by Pulmonologists Based on Cough Sounds

Cough sounds collected from patients who were confirmed to have pneumonia and those from other causes were heard by eight pulmonologists. The pulmonologists heard the same cough sounds from individual cases without any reference clinical information (e.g., sex, age, symptoms, or other clinical test results). They then determined whether the coughs were caused by pneumonia or another disease. They submitted their written responses for assessment.

In real medical scene, pulmonologists diagnose pneumonia based on evidence from various medical data, including not only cough sounds. Therefore, the most important index that can evaluate the performance of the algorithm is derived by the comparison with confirmed diagnoses of a pulmonologists. However, when diagnosing many respiratory diseases including pneumonia, the cough sound is a factor that clearly influences diagnosis by pulmonologists. Thus, this proposed analysis can sufficiently assist in confirming the necessity of algorithm using cough sound for pneumonia diagnosis.

### 2.6. Statistical Analysis

When analyzing the baseline characteristics of participants, continuous variables were captured as mean standard deviation (SD) or median interquartile range (IQR), depending on the normality, and each was tested with a t-test or Mann–Whitney U test. Normality was examined using Shapiro–Wilks test. Additional statistical analysis was performed to analyze the characteristics of the study population that participated in data collection and to visualize the probability of pneumonia for each case obtained as a result of the development algorithm to represent more clearly on the coordinate plane.

Categorical variables were described by number and percentage, and each was evaluated with Pearson’s chi-squared or Fisher’s exact test. STATA version 15.0 (StataCorp, College Station, TA, USA) was used as the statistical program to analyze the baseline characteristics. Results having two-sided *p*-values < 0.05 were considered to be statistically significant.

To visualize the data distribution of the pneumonia and non-pneumonia groups, the t-stochastic neighbor-embedding (t-SNE) visualization technique was used. Similarity among data in the space displaying the diagnostic results of the developed AI-based diagnostic procedure and those among data in the embedding space displaying the visualization results were defined. Moreover, direction of minimizing the Kullback–Leibler divergence was determined, whereby the data distribution in the pneumonia and non-pneumonia groups was visualized [38,39].

## 3. Results

### 3.1. Population

A total of 30 cough patients were enrolled. The mean age was 48.5 ± 13.8 years and 43.3% were males. Body-mass index (BMI) was 24.4 ± 4.6 $\mathrm{kg}/m^{2}$ and 3.3% were current smokers. Cough symptoms had occurred approximately 7 days prior to visiting the clinic (IQR: 7–15 days) and accompanying symptoms included sputum (93.3%), wheezing (46.7%) and breathing difficulty (46.7%).

There were no statistically significant differences of basic medical information including sex, age, BMI, smoking history, and cough severity between the pneumonia (*n* = 15) and non-pneumonia (*n* = 15) groups (Table 2). The non-pneumonia causes of cough included postinfectious cough, acute or chronic rhinosinusitis, acute or chronic bronchitis, bronchiectasis, asthma, eosinophilic bronchitis.

### 3.2. Characteristics Quantification with Psychoacoustics Aspect

Figure 2 shows the psychoacoustics and sound-energy characteristics found in the pneumonia and non-pneumonia groups. To compare the characteristics of pneumonia and non-pneumonia coughs in Figure 2, the single cough sound was divided into three phases by using vertical dotted lines. The first phase is 0.15 s interval from the start of cough, the second phase is 0.23 s interval from the first phase, and the third phase is last 0.12 s interval until the cough ends.

In Figure 2a,b, variation patterns in normalized sound level were analyzed for cough sounds in both groups. The pneumonia group showed instantaneously increasing level during the first phase immediately after the occurrence of cough, followed by a gradual decrease during the second phase. At the third phase, a louder sound than those from other disease group occurred. Such findings confirmed a pattern of a distinct increase in level at the end of the coughs. The non-pneumonia group showed a similar pattern with instantaneous increases in sound magnitude during the first phase. However, the sound pressure gradually decreased without a clear distinction between the second and third phases and dissipated.

These characteristics of cough sounds were observed at the loudness in the same way. Figure 2c,d show the variation pattern of the loudness. In the case of pneumonia cough, the value increased at the same time of cough occurrence, and gradually decreased with time. The pressure increased again in the second phase of the cough. The slope of the pressure variation was negative after the occurrence of cough and become positive again. Loudness increased 30 sone to 55 sone at the second local maximum. However, in the case of non-pneumonia, no other maximum was observed except for one local maximum value that appeared at the start of cough.

### 3.3. Sound-Energy Distribution

Figure 2e,f show the STFT of cough sounds for pneumonia and non-pneumonia groups. It represents the change of frequency component with the passage of time. Figure 2g,h show the energy ratio of cough sounds from each group derived through the EMD method. In the pneumonia group, the resonance frequency components ≥1000 Hz appeared alongside the occurrence of coughs, and the same component was constantly sustained until the third phase. Resonance components were found to be distributed and strengthened in the high-frequency range ≥1000 Hz due to loss of sound absorptions. As the energy distribution of cough occurrences appearing throughout the entire frequency range decreased, the energy ratio also decreased over a certain interval. Subsequently, the energy ratio increased again because of the high-frequency component that was consistently sustained during the second phase and thereafter. However, the energy ratio decreased during the third phase because of the energy distribution of the low-frequency component generated as the vocal cords closed, after which the cough dissipated. In other words, the local maximum value was shown at second phase of the cough due to the high frequency component distributed in the band of 1000 Hz or higher.

The specific frequency component sustained continuously was not observed in the non-pneumonia group. Thus, the division between cough phases was not clear, and the values were maintained at a relatively consistent level. In the quantitative aspect of the fluctuation range, the difference between maximum and minimum energy ratio values was 1.2 for pneumonia and 0.2 for non-pneumonia. Sometimes, there was a tendency for the values to decrease rapidly because of the strong energy distribution of the low-frequency component in the third phase for non-pneumonia groups.

### 3.4. Diagnostic Accuracy of AI-Based Algorithm

Reference data shown in Figure 3 were obtained via pre-processing in preparation for the application of data augmentation for loudness and energy ratio shown in Figure 2. Subsequently, the database constructed through the data augmentation process was used for reference-based training. Moreover, after constructing the LSTM-based machine-learning algorithm, it was applied to determine whether the cough was caused by pneumonia or non-pneumonia for testing performance. The diagnosis results showed 90.0% sensitivity, 78.6% specificity and 84.9% accuracy for new cough cases of pneumonia (*n* = 70) and for those of non-pneumonia (*n* = 56). Figure 4 shows the distribution of difference clusters for each case using the t-SNE visualization.

### 3.5. Comparison with Pulmonologists

The same cough sounds were used by eight pulmonologists for assessment. The results showed that their correct response rate was 56.4 ± 5.8% for the same cough cases, whereas sensitivity and specificity of 40.7 ± 15.6% and 71.4 ± 11.8%, respectively. The accuracy, sensitivity, and specificity value of developed AI-based algorithm and each pulmonologist were expressed together for the comparison in Table 3. In addition, a logistic regression model was constructed to plot the ROC curve based on these diagnosis results. In Figure 5, the ROC curves of AI and pulmonologists were represented in (a), and the AUC values obtained from each curve are plotted in (b). AUC value can be used for comparison of diagnostic performance [40] and AI was 0.84, pulmonologists averaged 0.56. It means the developed algorithm has higher performance than that of pulmonologists for diagnose pneumonia, based on cough sounds.

## 4. Discussion

The present study attempted to analyze the cough characteristics of adults and demonstrate that AI-based diagnostic procedure using loudness and energy ratio could be used to effectively diagnose pneumonia using cough sounds.

Cough sounds caused by pneumonia showed the following characteristics. Loudness distinctly increased in the latter phase of cough. In addition, the energy ratio increased by the continuous high-frequency component after the second phase. The loudness and energy ratio were selected as the objective characteristic indicators via the acoustic analysis of the cough sounds. Using these features, LSTM-based machine-learning algorithm was constructed for diagnosis. To relieve the problem of temporospatial constraints of establishing big datasets, the data augmentation technique that generates new data by accounting for characteristics of the data obtained by measurements was applied. This machine-learning algorithm showed 90.0% sensitivity and 78.6% specificity for 126 cough sounds that were not used for training. It also showed an overall accuracy of 84.9%, which confirmed that it can effectively diagnose pneumonia symptoms. Such diagnostic performance showed even higher than the results obtained by evaluation of cough sounds by pulmonologists. This means that the algorithm has significant reliability as an assistant index for the diagnosis of pneumonia.

Various studies have attempted to apply AI technology for classification of respiratory disease by cough sounds. However, the previous studies have been pioneering role by combining cough sound signals and AI technology to diagnose respiratory disease of pediatric patients. Personal respiratory health status changes because of the influence of countless environmental variables. For actual applications, robust symptom differentiation is required. By using the proposed unique acoustic features, the present study newly identified the characteristics of cough sounds for pneumonia and incorporated the findings into a machine-learning algorithm. The algorithm in the present study differentiated pneumonia and non-pneumonia cases with accuracy up to 80%.

The error cases were analyzed to examine the performance estimation results of developed algorithm. There were cases that some single coughs occurred in sequence within a short time interval. In this situation, the later the cough occurred, intensity of energy was not significantly large. Therefore, the characteristics of disease related to the distribution of frequency components were not clearly reflected to the cough. There were cases in which total energy distribution has been shifted relatively upward or downward according to the individual characteristics of the patient. This factor seems to be affectable about the diagnosis results of the algorithm. It is expected that such errors can be reduced by supplementing the data set by performing measurements on a larger number of patients through follow-up studies in the future.

Moreover, in this study, the new characteristic factors were suggested that effectively reflect the acoustic characteristics of pneumonia cough. By learning the algorithm with only these features and analyzing the performance evaluation results, it was possible to confirm the diagnostic contribution of proposed features. In follow-up studies, the diagnostic accuracy and robustness of the algorithm should be further enhanced by using additional characteristic factors including new acoustic factors, breathing sounds and medical characteristics.

The global COVID-19 pandemic has emphasized the importance of diagnostic techniques that minimize the active involvement of medical professionals and reduce the exposure of equipment while still upholding the rights of patients. Furthermore, the algorithm that relatively easily and simply diagnose the infection of the disease by cough sound would allow early detection of severe diseases. Our algorithm is also valuable as a diagnostic assistant technology that allows patient to check the possibility of infection in a simple and convenient way prior to detailed diagnosis by pulmonologists. For situations involving self-isolation, such as with COVID-19, far simpler methods are needed for continuous monitoring. In this respect, the present study has clinical significance, having presented the potential of an AI-based pneumonia diagnostic algorithm applications that listens to cough sounds. In addition, the developed algorithm is judged to be capable of sufficiently measuring and analyzing even with the microphone sensor built into general-purpose devices (e.g., mobile phones). Therefore, if the algorithm developed in the future is modularized into a form that can be operated in a new platform environment, applicability will be improved and contribute to telehealth implementation with quick screening.

The limitations of the present study include the following issues. To examine whether the sample size is appropriate, the sample size calculation formula required for medical research was applied as below [41].
(4)$$n=\frac{{\left[Z_{\frac{\alpha }{2}}\sqrt{P_{0}(1-P_{0})}+Z_{\beta }\sqrt{P_{1}(1-P_{1})}\right]}^{2}}{{\left(P_{1}-P_{0}\right)}^{2}},$$
where significance level *α* = 0.1, statistical power 1 − *β* = 0.8, $P_{0}\;$= 0.63, and $P_{1}=0.85$. The values of $P_{0}$ and $P_{1}$ were determined based on the results of a pneumonia diagnosis test through cough sound targeting pulmonologists. As a result of the calculation of the equation, n=24.73≈25, which can be judged to be adequate for the sample size in this study with a total of 30 patients. But as a single-center study in Korea and included only patients who visited a tertiary clinic, there was a limit to further increasing the confidence interval and power. To overcome the temporal and spatial constraints on data acquisition in this study, the number of data used for constructing AI algorithm has been increased by applying a data augmentation process. Due to the current global COVID-19 pandemic, it was difficult to collect additional patient data. The research team is planning a follow-up study in the future to secure additional data set. Additional studies are necessary to supplement and improve the algorithm in a larger sample population and generalize the findings in this study. The diagnosis of pneumonia was based on the decision by specialists, there might be possible discordance of the diagnosis of pneumonia among the physicians. In addition, comorbidities and smoking status could affect the results. However, these factors were not fully adjusted in the analysis due to a small number of cases. The present study applied the algorithm to adults. Further sound data collections required to determine if this method can be successfully applied to children for similar diagnosis. In addition, if the characteristic of each group is quantified by subdividing the age group and reflected in the cough characteristics, the applicability of developed algorithm can be further expanded. The loudness and energy ratio of cough sounds were believed to have no variability by different ethnic or language. Future studies are required to confirm whether the analytical method used in the present study is valid for multiple groups.

## Figures and Tables

**Figure 1 sensors-21-07036-f001:**
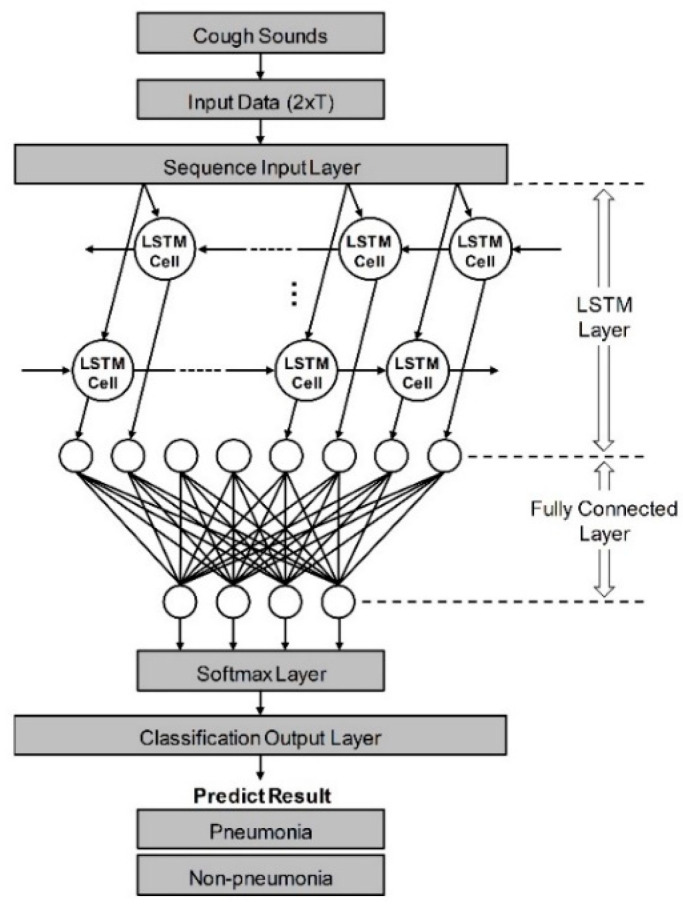
Schematic of LSTM-based machine-learning algorithm for differentiation of pneumonia. Input data from cough sounds include loudness and energy ratio. The algorithm consists of five layers: sequence input, LSTM, fully connected, softmax, and classification output layer.

**Figure 2 sensors-21-07036-f002:**
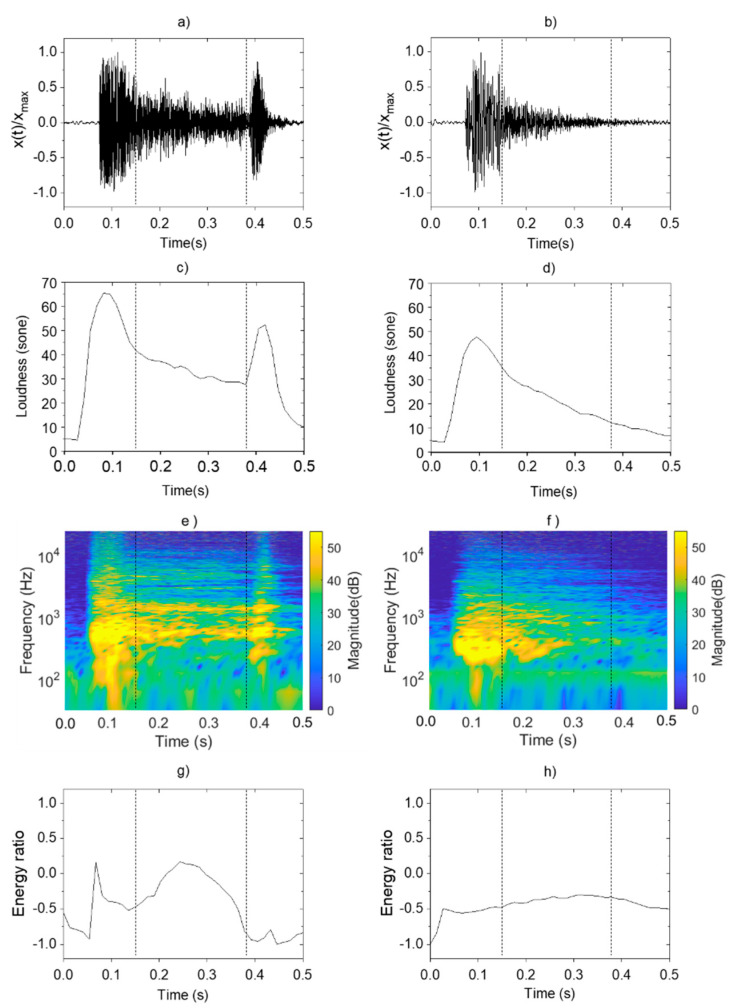
Analysis and comparison of acoustic characteristics of pneumonia and non-pneumonia cough sounds: sound pressure level of (**a**) pneumonia; (**b**) non-pneumonia; loudness of (**c**) pneumonia; (**d**) non-pneumonia; STFT of (**e**) pneumonia; (**f**) non-pneumonia; energy ratio of (**g**) pneumonia; (**h**) non-pneumonia.

**Figure 3 sensors-21-07036-f003:**
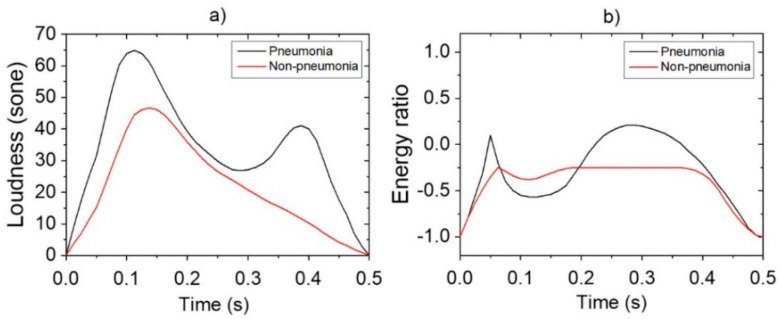
Pre-processing of features for data augmentation: (**a**) loudness; (**b**) energy ratio.

**Figure 4 sensors-21-07036-f004:**
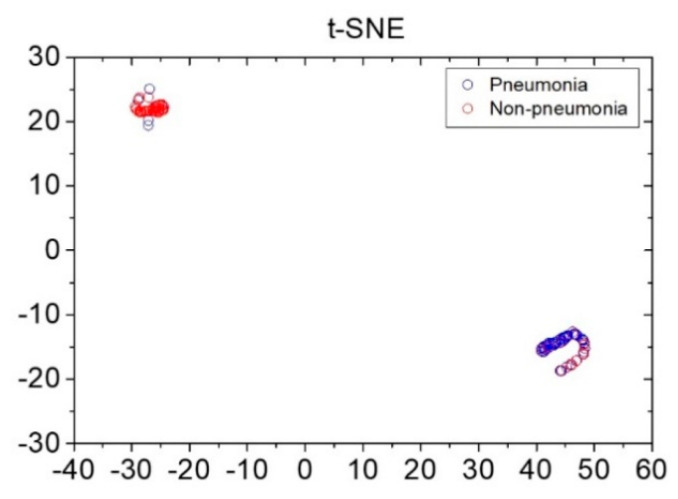
Distribution of pneumonia and non-pneumonia data clusters using the t-stochastic neighbor-embedding visualization technique. x and y axes represent dimensions 1 and 2, respectively, derived using the t-stochastic neighbor-embedding algorithm.

**Figure 5 sensors-21-07036-f005:**
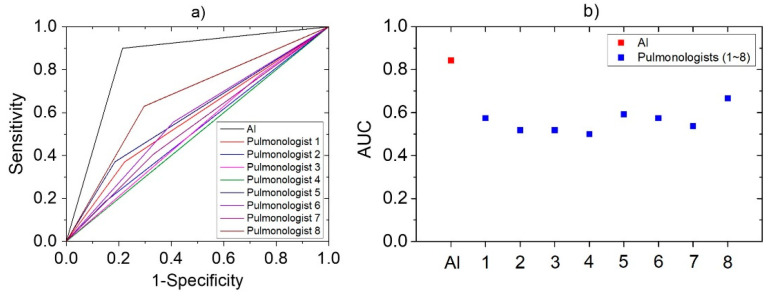
The statistical analysis derived by logistic regression models based on the pneumonia diagnostic results of AI and pulmonologists: (**a**) ROC Curve; (**b**) AUC.

**Table 1 sensors-21-07036-t001:** Technical specifications of the instrumentation.

PCB Piezoelectronics, 378B02	Nominal Microphone Diameter	12 mm
	Frequency Response Characteristic (at 0° incidence)	Free-Field
	Sensitivity	50 mV/Pa
	Frequency Range (±2 dB)	3.75 to 20,000 Hz
	Temperature Range (Operating)	−40 to +80 °C
	Static Pressure Coefficient	−0.013 dB/kPa
	Excitation Voltage	2 to 20 mA
Recorder Settings	Sampling frequency	48,000 Hz
	Sampling time (Δt)	0.020833 ms
	Coupling type	ICP

**Table 2 sensors-21-07036-t002:** Descriptive characteristics of the study population. Continuous values are presented as mean ± SD, and categorical variables are presented as number (%). Cough severity was measured using visual analogue scale from 0 to 10 (highest).

	Pneumonia (*n* = 15)	Non-Pneumonia (*n* = 15)	*p* Value
Age, years	49.7 ± 14.4	45.9 ± 13.4	0.469
Male	6 (40%)	9 (60%)	0.465
BMI, kg/m^2^	25.5 ± 4.9	23.5 ± 4.2	0.258
Smoking			0.141
Never	7 (46.7%)	12 (80.0%)	
Ex-smoker	7 (46.7%)	3 (20.0%)	
Current smoker	1 (6.7%)	0 (0.0%)	
Visual analogue scale of cough	7.1 ± 2.2	7.2 ± 2.2	0.935
Symptoms			
Sputum	15 (100.0%)	13 (86.7%)	0.4642
Wheeze	6 (40.0%)	8 (53.3%)	0.7144
Dyspnea	8 (53.3%)	6 (40.0%)	0.7144

**Table 3 sensors-21-07036-t003:** Comparison of pneumonia diagnostic results between artificial intelligence-based algorithm and pulmonologists using cough sounds.

		Accuracy (%)	Sensitivity (%)	Specificity (%)
AI		84.9	90.0	78.6
Pulmonologist	1	58.2	37.0	78.6
	2	52.7	18.5	85.7
	3	49.1	22.2	75.0
	4	50.9	51.9	50.0
	5	60.0	37.0	82.1
	6	58.2	55.6	60.7
	7	54.6	40.7	67.9
	8	67.3	63.0	71.4

## Data Availability

Not Applicable.

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
