# Peer review of "Diagnosis of Pneumonia by Cough Sounds Analyzed with Statistical Features and AI"

_sensors, 2021, doi:10.3390/s21217036_

Round 1
Reviewer 1 Report
This is an exciting area of investigation with important clinical relevance. For that reason, it is important the the authors address 3 major weaknesses.
1. The manuscript needs a power calculation to show if the sample size is adequate. With such a small sample size there is the possibility of a type I error. The authors must describe how the sample size was calculated
2. The construction of the AI appears weak, or needs much mor explanation. Only 4 patients were used. What was done to guard against overfitting? The authors need to justify why data from only four patients is adequate to produce the algorithm
The comparison to pulmonologists analyzing cough sounds only is probably not relevant. I’d omit this as it has no real world relevance. More important is the comparison with the confirmed diagnoses.
Reviewer 2 Report
In this paper the authors propose a diagnosis method for pneumonia by cough sounds analyzed with statistical feature and AI.
Section 1must be improved. You should introduce the problem in more detail so that the reader is immediately clear about the purpose of your study. Specify better the essential elements of the problem. You should add more information in the introductory part, you should add other works that have also addressed the problem. You must properly introduce your work, specify well what were the goals you set yourself and how you approached the problem. At the end of the section, add an outline of the rest of the paper, in this way the reader will be introduced to the content of the following sections.
Section 2 must be improved. Describe the population precisely (age, sex, type of work, place where they live, other diseases, etc). This way you will be able to demonstrate that the sample is representative of the population, and you will also be able to obtain other useful information about the disease. You also need to better describe the descriptors you extracted from the data and how you will use them in the model. You also need to properly introduce the Machine Learning technology you will be using (LSTM). It also justifies your choice (Why did you choose LSTM?). In the end, clarify what these statistical analyzes are for, because it is not clear. Will you use these descriptors as model input?
Section 3 must be improved. You need to spend more time presenting the results. It is okay to start with the population sample under investigation. Then you can move on to an exploratory analysis of the data and then finally present the results of the classification with LSTM.
Section 4 must be improved. This section needs to be moved to the introduction, or to the methodology. This is the final section, here you have to summarize what you have achieved in your work, not what others have done. The next part should also be moved.
65-85) In this section you present the sample you have analyzed. This is your data so you need to be more detailed, describe the population precisely (age, sex, type of work, place where they live, other diseases, etc). This way you will be able to demonstrate that the sample is representative of the population, and you will also be able to obtain other useful information about the disease.
95) How were the 30 patients selected? How many for each class (pneumonia, not pneumonia)?
85-96) 126 records are few to train an algorithm based on Machine Learning.
106-107)” After that, to quantify the acoustic features, loudness was analyzed from a psychological aspect”. Talk about this sentence, in what sense were the data analyzed from the psychological aspect? Maybe you meant using psychoacoustic descriptors?
110) Do not use abbreviation such as i.e.
106-110) Introduce adequately the topic (Loudness)
124)” was derived from the frame at 0.05 s” Add references to support this choice.
99-128) At the end of this section, summarize the acoustic descriptors you will use as input for your model.
130)Why you have selected only 4 participants? You had 30 patients available, you could have used 80% for the training and the rest for validation and testing.
134-144) Explain better how you performed the data augmentation. First explain what it consists of and then describe in detail how you carried out this operation.
153-158) Introduce adequately the LSTM. Start by introducing deep learning and then list papers that have used this technology for sound event detection. For example:
- Hayashi, T., Watanabe, S., Toda, T., Hori, T., Le Roux, J., & Takeda, K. (2017). Duration-controlled LSTM for polyphonic sound event detection. IEEE/ACM Transactions on Audio, Speech, and Language Processing, 25(11), 2059-2070.
- Li, J., Dai, W., Metze, F., Qu, S., & Das, S. (2017, March). A comparison of deep learning methods for environmental sound detection. In 2017 IEEE International Conference on Acoustics, Speech and Signal Processing (ICASSP)(pp. 126-130). IEEE.
- Ciaburro, G., & Iannace, G. (2020, September). Improving smart cities safety using sound events detection based on deep neural network algorithms. In Informatics(Vol. 7, No. 3, p. 23). Multidisciplinary Digital Publishing Institute.
180-195) Clarify what these statistical analyzes are for, because it is not clear. Will you use these descriptors as model input?
212-213) Figure 2 shows a spectrogram, you should introduce it properly otherwise non-expert readers will not be able to understand its content.
254) Figure 2 contains 8 subplots which must all be labeled. Once this is done, in the caption you will have to describe in detail what they contain one by one. Finally it is necessary to add the unit of measurement to the colored map of the spectrogram.
308-326) This section needs to be moved to the introduction, or to the methodology. This is the final section, here you have to summarize what you have achieved in your work, not what others have done. The next part should also be moved.
Reviewer 3 Report
On the base of my experience in biomedical sounds analysis, while I appreciate this work very much, nevertheless what it is missing is (at least) a tentative to propose a model for the cough sounds and a model for the cough spectral content. I feel that the frequency content of the cough sound will also depend on the "voice sound" of the subject as it is very well known that the identity of a person can be detected by his/her cough besides his/her voice. I mean that as we can recognize a person from his/her voice, the same we can recognize a person from his/her cough. Thus I wonder if the authors have considered this aspect. Apart from this I like this paper and I feel it merits being published also because results show that ROC curve of the AI algorithm is much better of the (tossing coin like) ROC curve of doctors.
Round 2
Reviewer 1 Report
The authors have not addressed the concerns I raised in the first review.
Reviewer 2 Report
The authors addressed all the reviewer's comments with sufficient attention and modified the paper consistently with the suggestions provided. The new version of the paper has improved significantly both in the presentation that is now much more accessible even by a reader not expert in the sector, and in the contents that now appear much more incisive. The addition of a sufficient reference bibliography gave consistency to the authors' statements and the results they achieved.
Minor revision:
41)” diagnoses[2]” Leave a space between the word and the reference, I have seen that this problem is present throughout the paper. Correct all occurrences.
130) Add a table with technical specifications of the instrumentation used (HEAD acoustics GmbH, SQobold and PCB Pie- 110 zotronics, 378B02). You could also add images of the equipment in place.
157) I had already reported this criticality, unfortunately your answer did not convince me. When an AI-based algorithm is developed, the main part of the data is used for training the model. This is because it is in this phase that it is necessary to provide the algorithm with the greatest amount of information to derive the characteristics of the system. Spitting usually takes place in this form: 70% for training, 15% for validation, and the remaining 15% for testing. Leaving most of the data for testing is a mistake. Try to better justify your choice or try to delve into the model with more data.
182-183) Add references to support these statements.
194) Explains in detail the role of the 5 layers present in Figure 1: sequence input, LSTM, fully connected, softmax, and classification output layer.
264) It’s better In Figures 2 a) and 2 b)
274) It’s better In Figures 2 c) and 2 d)
282) It’s better In Figures 2 e) and 2 f)
284) It’s better In Figures 2 g) and 2 h)
